# GuardFormer 🛡: Guardrail Instruction Pretraining for Efficient SafeGuarding

**This paper contains examples of harmful language. Reader discretion is recommended.**

**James O' Neill**[†,1]     **Santhosh Subramanian**[†,1]     **Eric Lin**[1]     **Abishek Satish**[1]

james@dynamofl.com

**Vaikkunth Mugunthan**[1]

## Abstract

Large language models (LLMs) have shown promise in guardrailing against undesired behaviors, but their high inference costs, memory consumption, and unstructured outputs can be prohibitive. In this work we propose guardrail-specific instruction pretraining using a synthetic data generation pipeline. The data generation process is tailored towards generating policies that define the scope of the guardrail, compliant and non-compliant prompts, rationales when non-compliant and the output binary compliant or non-compliant label. From this, we propose a new guardrail model called `Guardformer` and show when further few-shot fine-tuned it significantly outperforms current state of the art (SoTA) while only requiring 512MB in storage. `GuardFormer` is orders of magnitude smaller than baselines such as `gpt-4`, yet significantly outperforms it while having the ability to learn from multiple custom policies at once.

Empirical evaluation across 7 public datasets and 4 novel guardrail benchmarks demonstrates our efficient classifiers' superiority over state-of-the-art LLMs and third-party APIs. Our models achieve average F1 score improvements of **29.64** and **21.07** points compared to Aegis-LlamaGuard and `gpt-4o`, respectively, in distinguishing safe from unsafe behaviors. Notably, models trained on our synthetic data consistently outperform those trained on real data, even when evaluated against custom-defined guardrailing policies, underscoring the efficacy of our approach.

## 1 Introduction

The widespread use of large language models (LLMs) in both the public and private domains has led to an increasing concern around guardrailing against malicious prompts [Biswas and Talukdar, 2023, Greshake et al., 2023, Manczak et al., 2024]. While there has been a concerted effort to defend against misuse of LLMs, current guardrailing and safety alignment approaches can lead to considerable performance degradation on safe and non-malicious prompts, reducing the models general capabilities [Qi et al., 2023, Jain et al., 2023]. In contrast, guardrails that are independent of the main LLM being used avoid this issue of safety alignment degrading generalization performance. While 3[rd] party API services and publicly available models (e.g PromptGuard and LlamaGuard [Inan et al., 2023]) offer

---

[1]Dynamo AI, San Francisco, California, United States of America

[†]These are the leading authors and contributors of this paper.

38th Conference on Neural Information Processing Systems (NeurIPS 2024).

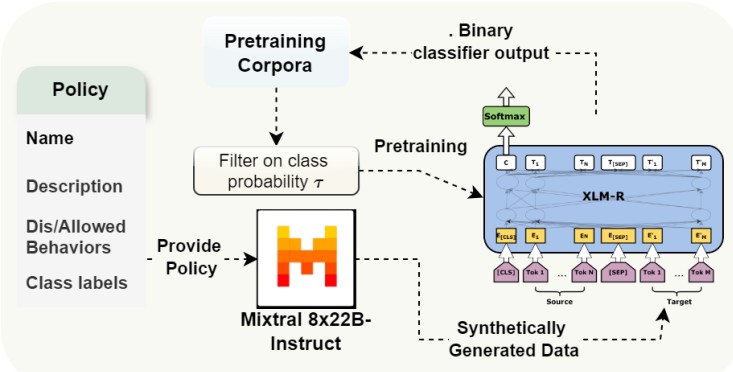

Figure 1: **GuardFormer**: Creating Robust Guardrails with guardrail-instruction pretraining and guardrail classification using Synthetic Guardrail Data Generation.

different solutions to this issue of guardrailing while not diminishing the LLMs general capabilities, they are limited in generalization performance, inference speed and adaptability (i.e transfer learning is difficult without retraining).

In this paper, we show that through the use of a well crafted synthetic data generation pipeline that our robustly fine-tuned classifiers can significantly outperform current state of the art (SoTA) while being orders of magnitude smaller w.r.t the number of model parameters. This involves describing each task with task definitions that include a concise summary of the task, allowed and disallowed behaviors and examples of safe and unsafe behaviors. We demonstrate the effectiveness of these classifiers on various safety, toxicity and prompt injection public benchmarks and show significant improvements over LLamaGuard-[1,2,3]-7b [Inan et al., 2023], Nemo Guardrails [Rebedea et al., 2023], Azure Content Safety, GPT-3.5-turbo/4/4o [OpenAI, 2023a], Meta PromptGuard [Inan et al., 2023] and OpenAIs Content Moderation API [OpenAI, 2023b]. One challenge to this initial approach is that we require a fine-tuned classifier for each domain for optimal performance. To address this challenge we further show that it is possible to maintain better performance over the aforementioned baseline using a unified guardrail that performs guardrail instruction pretraining learning by unifying synthetically generated policy-specific datasets. Below we summarize our contributions:

- Guardrail classifiers that are 14 times faster than the best performing LLM (`gpt-4`) while outperforming it on public datasets by 21.07 F1 and 5.13 F1 on our proposed `CustomGuardBenchmark`.
- A multi-task learning approach to guardrailing, we refer to as `GuardFormer` that outperforms a single-task guardrailing model, referred to as `PolicyGuard` by performing guardrail specific pretraining on synthetic data.
- A synthetic data generation pipeline for guardrailing, for both classifier pretraining for generalizability and policy specific fine-tuning for task specificity. This includes a self-reflection step that improves the label judgements by reflecting on the LLMs intial label prediction.
- An analysis of how guardrail performance varies as a function of 1) the number of training samples used for training, 2) training on synthetic or real data and 3) the number of active fine-tuning parameters required with and without pretraining.

## 2   Related work

Below describes work related to the main aspects of this work.

**Generative Content moderation.** Ensuring safety has been an active area of research for several years. Most recently, LLMs have been used solely for guardrailing. This includes LlamaGuard-7B/2-8B/3-8B [Inan et al., 2023] have defined policy descriptions and unsafe categories that outline what 'Can' and 'Should not' be allowed for input prompts and output responses. Ghosh et al. [2024] have built upon LlamaGuard-2 by using safety-based adapter fine-tuning of LlamaGuard 2 on the Aegis Safety dataset that outlines a broad taxonomy of

13 critical safety risk categories. Nemo Guardrails Rebedea et al. [2023] have introduced `programmable guardrails` whereby a specialized modeling language, Colang, can be used to define behaviors by giving either behavioral definitions or examples of such in a programmatic manner. While these LLMs have shown promise, it still remains infeasible to run (7B or larger) models for many latency (or memory) critical applications. Additionally, next token prediction in these generative models are not guaranteed to predict the defined safe or unsafe categories which potentially makes output parsing unreliable and difficult.

**Discriminative Content Moderation** Bert-based classifiers have been used to detect offensive or toxic inputs [Vidgen et al., 2020, Deng et al., 2022]. More more recent work has focused on the use of LLMs through APIs such as Perspective API [Lees et al., 2022], OpenAI Content Moderation API [Markov et al., 2023] (categories including toxicity, threat, harassment, and violence) and Azure Content Safety API [Microsoft, 2023] (categories include hate and violence) that provide a severity score between 0-6. While bert-based classifiers have the benefit of being much smaller than current LLMs, to date they have lacked the necessary training data to be robust against guardrail domains and topics of interest. Lee et al. [2024] have also focused on training binary classifiers to guard against unsafe prompts and responses. They use affirmative prefixes to encourage safety-aligned LLMs to generate instructions, responses and rely on LlamaGuard-3 to generate corresponding labels. In contrast, to our work, we are not limited to a single jailbreaking technique to generate text from LLMs, we do not need to rely on an existing seed dataset (a policy definition instead), we only require one LLM generator (they use 3 different LLMs) and our pretraining dataset is an order magnitude larger (>1 million training samples, compared to their 100k training samples).

Our work addresses shortcomings of these prior works.

## 3 Methodology

Below we describe how we synthetically generate safe and unsafe samples and refine policy definitions for improved generation on various guardrail tasks. We then describe the model pretraining, fine-tuning and model merging process.

### 3.1 Synthetic Data Generation

For Synthetic Data Generation (SDG), we begin by defining a description of the task, which we refer to as a policy $\mathcal{P}$. Here, $\mathcal{P}$ includes a policy name $\mathcal{P}_{\text{name}}$, description $\mathcal{P}_{\text{desc}}$, allowed behaviors $\mathcal{P}_{\text{allowed}}$, disallowed behaviors $\mathcal{P}_{\text{disallowed}}$ and an optional $\mathcal{P}_{\text{examples}}$ that gives examples of safe and unsafe prompts. Given $\mathcal{P}_{\text{disallowed}}$, a seed dataset $\mathcal{D}_{\text{seed}} := \{(x^i_{\text{safe}}, r^i_{\text{safe}}, y^i_{\text{safe}})\}^{N_{\text{safe}}}_{i=1} \bigcup \{(x^i_{\text{unsafe}}, r^i_{\text{unsafe}} y^j_{\text{unsafe}})\}^{M_{\text{unsafe}}}_{j=1}$ is generated where $x_{\text{safe}}$, $r_{\text{safe}}$ and $y_{\text{safe}}$ are a compliant prompt, a rationale for compliancy and label and $x_{\text{unsafe}}$, $r_{\text{unsafe}}$ and $y_{\text{unsafe}}$ are a noncompliant prompt, a rationale for noncompliancy and label respectively. We can formulate the SDG process as a conditional distribution $p(\mathcal{D}|\mathcal{P};\mathcal{G})$ where $\mathcal{G}$ is the LLM data generator. Once $\mathcal{D}$ is generated, we refine the policy to improve clarity using a prompt template that prompts $\mathcal{G}$ to self-reflect on its own label judgements for all $y_{\text{unsafe}}$ and $y_{\text{safe}}$ with the aim of recorrecting any incorrectly generated prompts. For our public benchmarks that contain training datasets along with test sets that are used for benchmarking (e.g BeaverTails [Ji et al., 2024]), a set of example unsafe inputs in $\mathcal{P}_{\text{examples}}$ are used to bias the data generation towards prompts that are within the same domain.

### 3.2 Custom Policy Guardrailing

Given the synthetic data generation process described by $p(\mathcal{D}|\mathcal{P};\mathcal{G})$, we first train policy-specific fine-tuned classifiers, known as `PolicyGuard`. This method uses the generated dataset $\mathcal{D}$ to create highly specialized models capable of accurately identifying policy violations across diverse domains. Let $f_\theta$ denote our base classifier with parameters $\theta$, which can be instantiated as large pre-trained language models (e.g RoBERTa-large). We fine-tune $\{_\theta$ to create a policy-specific classifier $f_{\theta_\mathcal{P}}$ that maximizes performance on the task defined by policy $\mathcal{P}$. We formulate the fine-tuning objective as $\mathcal{L}_{\text{CE}}(\theta) = -\frac{1}{|\mathcal{D}|} \sum_{(x_i, y_i) \in \mathcal{D}} [y_i \log(f_\theta(x_i)) + (1 - y_i) \log(1 - f_\theta(x_i))]$ where $(x_i, y_i)$ are the prompt-label pairs from the synthetic dataset $\mathcal{D}$, and $f_\theta(x_i)$ is the predicted probability of the input being non-compliant with the policy. By

minimizing $\mathcal{L}_{\text{CE}}(\theta)$ the classifier learns specific nuances of $\mathcal{P}$ as represented in the generated data.

## 3.3 Multi-Policy Guardrailing

In contrast to `PolicyGuard`, in this section we describe `GuardFormer`, a novel approach designed to create a single, versatile model capable of performing well across all policies or tasks of interest. This approach not only enhances efficiency but also enables cross-task learning, potentially improving performance on individual tasks through shared representations.

To achieve this, we concatenate the SDG training datasets for all policies $\mathcal{P}_1, \mathcal{P}_2, ..., \mathcal{P}_N$, creating a unified dataset $\mathcal{D}_{\text{unified}}$ where each all $\mathcal{P}$ have been humanly created by a domain expert. We then use $\mathcal{D}_{\text{unified}}$ as a seed dataset to generate more diverse *synthetic* policies $\mathcal{P}'$ given the prompts and rationales in $\mathcal{D}_{\text{unified}}$. Then, with $\mathcal{P}'$ we prompt the generator $\mathcal{G}$ (e.g Mixtral 8x22B-Instruct) to generate safe and unsafe prompts and unsafe rationales, where applicable, given these new policies. This results in the full pretraining dataset $\mathcal{D}_*$ that consists of the original seed $\mathcal{D}_{\text{unified}}$ and the new diverse subset $\mathcal{D}_{\text{diverse}}$.

For each sample, we construct an instruction input that combines the policy description, prompt, and rationale. Formally, for a policy $\mathcal{P}_i$, a sample in $\mathcal{D}_*$ is represented as $\bar{x}_i = \mathcal{P}_{(i,\text{desc})}\backslash$n Query: $x_i$ [SEP] $r_i$ where $p_i$ is the prompt, $r_i$ is the corresponding generated rationale, and [SEP] is a separator token. We then train a multi-task model $f_{\theta_{\text{multi}}}$ on $\mathcal{D}_*$ by minimizing a combination of masked language modeling (MLM) loss, Alice$_{++}$ loss [Pereira et al., 2021] and classification loss:

$$\mathcal{L}(\theta_{\text{multi}}) = \lambda_1 \mathcal{L}_{\text{MLM}}(\theta_{\text{multi}}) + \lambda_2 \mathcal{L}_{\text{Alice}_{++}}(\theta_{\text{multi}}) + \lambda_3 \mathcal{L}_{\text{CE}}(\theta_{\text{multi}}) \tag{1}$$

where $\lambda_{1...3}$ are hyperparameters balancing the three loss components. The MLM loss $\mathcal{L}_{\text{MLM}}$ is defined as $\mathcal{L}_{\text{MLM}}(\theta_{\text{multi}}) = -\frac{1}{|\mathcal{M}|} \sum_{m \in \mathcal{M}} \log p(\bar{x}_m | x_{\backslash m}; \theta_{\text{multi}})$ where $\mathcal{M}$ is the set of masked tokens, $\bar{x}_m$ is a masked token, and $x_{\backslash m}$ represents the input with masked tokens. The Alice$_{++}$ loss $\mathcal{L}_{\text{Alice}_{++}}$ is designed to improve the model's robustness and generalization across tasks.

It is defined as $\mathcal{L}_{\text{Alice}_{++}}(\theta_{\text{multi}}) = \mathcal{L}_{\text{label}} + \alpha \mathcal{L}_{\text{virtual}}$ where $\mathcal{L}_{\text{label}}$ is the loss computed using gold labels and $\mathcal{L}_{\text{virtual}}$ is the virtual adversarial training (VAT) loss. The VAT loss is then defined as: $\mathcal{L}_{\text{virtual}}(\theta_{\text{multi}}) = \mathbb{E}_{x \sim \mathcal{D}} \left[ \max \delta : |\delta| \leq \epsilon \text{KL}\left(p(y|x; \hat{\theta}_{\text{multi}}) | p(y|x+\delta; \theta_{\text{multi}})\right) \right]$ where $\delta$ is a small perturbation bounded by $\epsilon$ and KL is the Kullback-Leibler divergence between the model's predictions for the original and perturbed inputs. This encourages consistent predictions under small input perturbations.

During inference, given a new input $x_{\text{new}}$ for a specific policy $\mathcal{P}_j$, we construct the instruction input as described earlier and use the trained model to predict: $y_{\text{pred}} = \text{argmax}_{y \in \{\text{safe, unsafe}\}} f_{\theta_{\text{multi}}}(x_{\text{new}})$. This guardrail instruction-based pretraining (GIP) allows the model to distinguish between different policies during both training and inference, effectively learning to handle multiple tasks within a single architecture while benefiting from shared representations across tasks.

# 4 Experimental Setup

## 4.1 Dataset Details

For our experiments, we split our fine-tuning and evaluation using synthetic data and compare to fine-tuning on real data on the training dataset from the public benchmark or if there is no training dataset for the public test dataset, we train on real related data (i.e the same domain). For our private benchmark, all our results for `PolicyGuard` and `MultiPolicyGuard` are fine-tuned on synthetic data. In the appendix we describe each policy description we use for both our public and private training datasets.

**Public Benchmarks** We first benchmark against publicly available datasets that are all available on the huggingface dataset hub[1], which we now provide their hub names. This includes 2 prompt-injection datasets (`deepset/prompt-injections` and

---

[1] https://huggingface.co/datasets

`xTRam1/safe-guard-prompt-injection`), 3 toxicity-based datasets ("toxicchat0124" from `lmsys/toxic-chat` Lin et al. [2023] and `SetFit/toxic_conversations_50k`) and 3 content safety datasets (`nvidia/Aegis-AI-Content-Safety-Dataset-1.0`, `mmathys/openai-moderation-api-evaluation` and `PKU-Alignment/BeaverTails`). Each datasets test set is converted into binary labels (safe/unsafe) where necessary (e.g openai-moderation).

**Private Benchmarking**   We also test our proposed guardrails on a private benchmark `CustomGuardBench`[2], which consists of datasets we refer to as `Safety`, `Finance`, `Tax` and `Injection`. These 4 datasets cover the prohibiting of unsafe discussions, financial advice, tax advice and prompt injection respectively. For all of these datasets, an expert compliance officer and policy informed annotators manually annotate the benchmark dataset given an aforementioned policy definition for each one.

## 4.2   Model Details

**Baseline Models.**   For 3[rd] party API services we use 1) OpenAI GPT models such as `gpt-3.5-turbo`, `gpt-4` and `gpt-4o` [OpenAI, 2023a]) OpenAI Content Moderation [OpenAI, 2023b], 3) Azure Content Safety and 4) Nemo Guardrails using `gpt-4o` as the generator. For the GPT-models we use batch completion through `litelllm`[3] library to optimally reduce API call response time. For our public available SoTA LLMs, we use LlamaGuard-1/2/3 [Inan et al., 2023], `Meta-Llama-3.1-8B-Instruct` [Dubey et al., 2024], `nvidia/Aegis-AI-LlamaGuard` [Ghosh et al., 2024] and Prompt-Guard-86M [AI, 2023].

**Finetuning Setup.**   The base models used for our experiments in finetuning and benchmarking `PolicyGuard` and `Guardformer` are RoBERTa$_{Large}$ Liu et al. [2019] and an instruction pretrained version of XLM-RoBERTa$_{Large}$ Wang et al. [2024]. The former is a standard well-established masked monolingual language model (MLM) model, while the latter is a multilingual MLM that has been trained from instructions to produce high quality sentence embeddings.

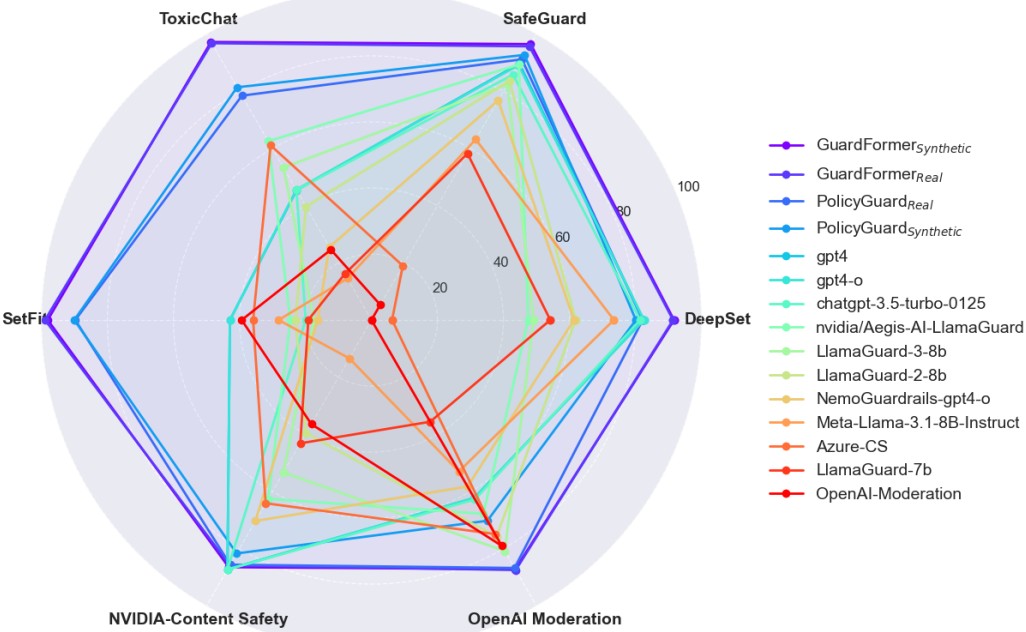

Figure 2: F1 Score Across `DeepSet`, `SafeGuard`, `ToxicChat`, `SetFit`, `Nvidia-Content-Safety` and `OpenAI Moderation` test datasets.

---

[2]This will be made public before the end of the year.

[3]https://github.com/BerriAI/litellm

# 5 Results

**Public Benchmarking**  Figure 2 shows the F1 scores per task on our curated public benchmark where the base model used for our proposed models is based on XLM-RoBERTa$_{\text{Large}}$. Overall, we find superior performance across a diverse set of toxicity, safety and prompt injection based tasks. `GuardFormer` consistently outperforms task-specific `PolicyGuard` models in both cases where we fine-tune on our synthetically generated training data (i.e Synthetic) and on the real training data (i.e Real). Most notably, `PolicyGuard` and `GuardFormer` significantly outperform both $3^{\text{rd}}$ party and publicly available LLMs. For example, `gpt-4o`, the best performing LLMs of our baselines, achieves 21.62 average F1 score points below our best performing guardrail model.

Figure 3 shows the overall guardrail performance for our proposed `PolicyGuard` and `GuardFormer` models and the baselines. We find the average F1 score for `GuardFormer` ranks the highest and outperforms these baselines, including `gpt4` and LlamaGuard models, by a large margin. Generally, `gpt4` and `gpt4-o` outperform publicly available LLM baselines that have been pretrained specifically for safety and other related policies. Of the LLamaGuard suite of models, we find `nvidia/Aegis-AI-LlamaGuard` outperforms the original Llama-Guard models. We also note while `OpenAI-Moderation` performs well on its own `OpenAI Moderation` dataset, it generalizes poorly to other safety and toxicity test sets. Lastly, we find that fine-tuning on our synthetic data instead of real public training data increases the performance. Table 1 shows the results when using `RoBERTa`$_{\text{Large}}$ as the base model, which unlike XLM-RoBERTa$_{\text{Large}}$has not been pretrained specifically for high performing sentence embeddings, nor has it been further pretrained with an instruct-based corpus. Due to this we see a drop in performance, however, we are still within 0.56 average F1 score points compared to 69.41 F1 obtained by `gpt-4` in Figure 2.

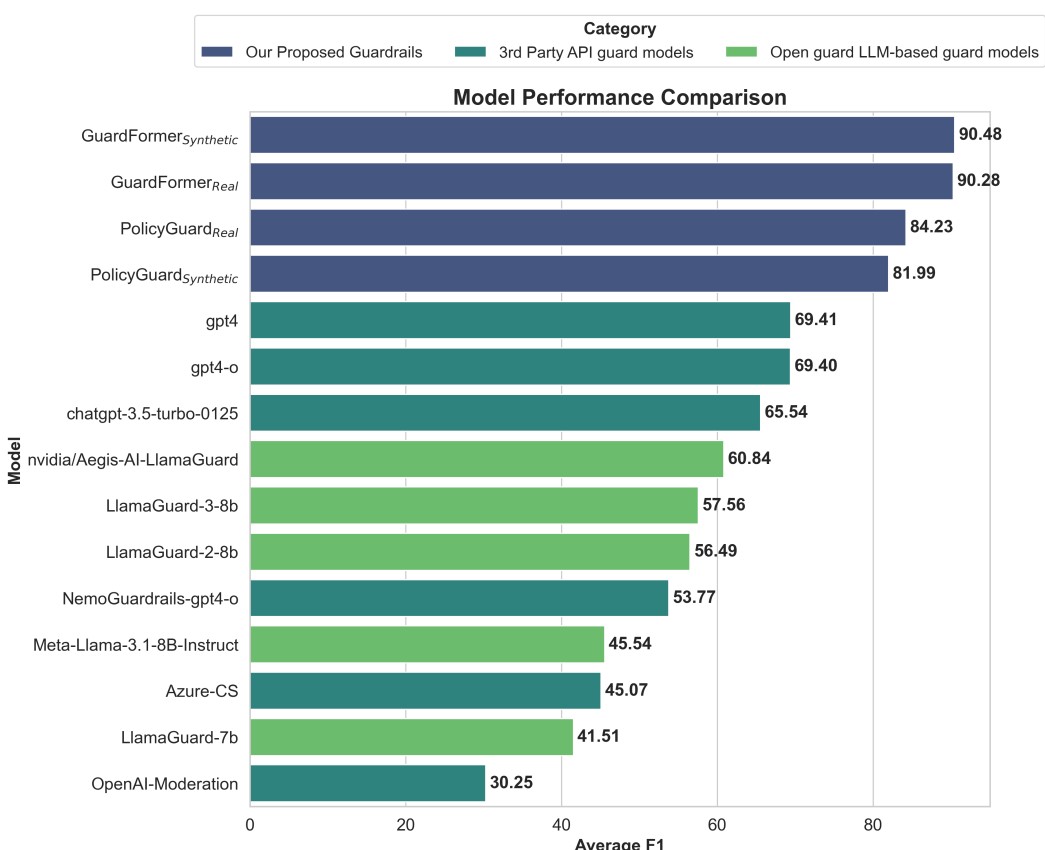

Figure 3: Average F1 Score Across 7 safety, toxicity and prompt injection test sets.

| Models | Score | Injection | | Toxicity | | Content Safety | |
|---|---|---|---|---|---|---|---|
| | | DeepSet | SafeGuard | ToxicChat | SetFit | NVIDIA-CS | Beavertails |
| PolicyGuard$_{\text{Synthetic}}$ | 57.89 | 56.81 | 81.31 | 36.54 | 15.99 | 80.87 | 75.85 |
| GuardFormer$_{\text{Synthetic}}$ | 67.97 | 63.06 | 86.04 | 56.73 | 35.82 | 83.93 | 82.28 |
| PolicyGuard$_{\text{Real}}$ | 56.54 | 57.92 | 79.65 | 34.81 | 15.23 | 78.54 | 73.12 |
| GuardFormer$_{\text{Real}}$ | 63.14 | 56.43 | 81.76 | 54.89 | 25.17 | 81.95 | 78.63 |

Table 1: **Comparing synthetic vs real training data with RoBERTA$_{\textbf{Large}}$.**

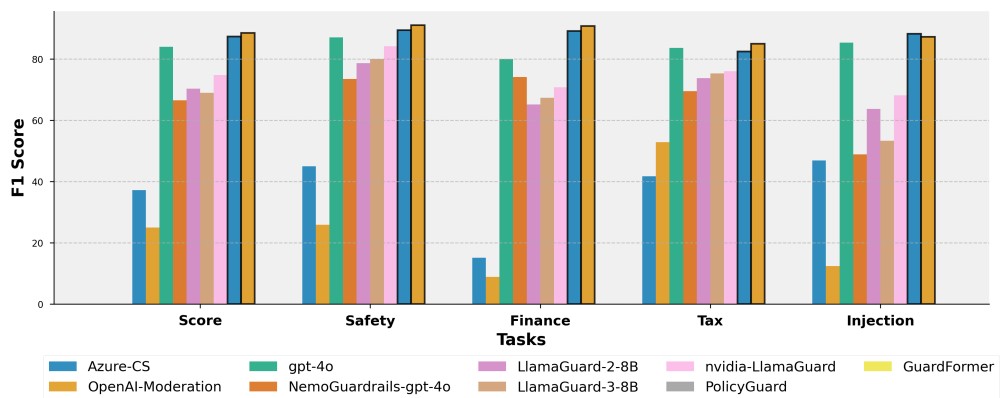

Figure 4: **Private benchmark results on DynamoGuardBenchmark**

Moreover, all other baselines are outperformed and significant improvements are found when using our synthetic training data compared to the real data training data that is available from each public dataset. Additionally, `GuardFormer` consistently outperforms `PolicyGuard` as we posit the effects of GIP in `GuardFormer` has more impact than XLM-RoBERTa$_{\text{Large}}$since it has not been pretrained with instructions prior to GIP.

**Private Benchmark Results** Based on the results presented in Figure 4, we find that `GuardFormer` demonstrates superior performance across all categories of the `CustomGuardBenchmark`[4]. `GuardFormer`s performance is particularly noteworthy in the Safety and Injection categories, where it achieves the highest scores of 91.83 and 88.62, respectively. While `gpt-4` is competitive in performance for safety and prompt injection, it suffers in performance on more specialized guardrail tasks, namely in `Finance` (i.e prohibiting financial advice) and to a lesser extent `Tax` (i.e Avoiding Tax Advice).

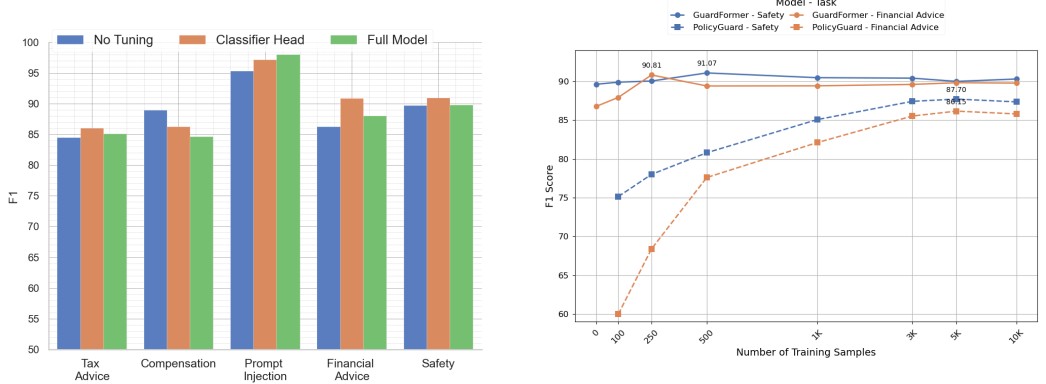

(a) `GuardFormer` - No Tuning vs. CFT vs. FFT

(b) `PolicyGuard` & `GuardFormer` **Learning Curves** on `Safety` and `Finance` test sets.

Figure 5: **Performance comparisons of different model configurations**

---

[4] `CustomGuardBenchmark` will be made public at https://huggingface.co/datasets

**GuardFormer converges faster during fine-tuning** Our analysis reveals that `GuardFormer` achieves optimal performance with less active parameters tuned during task-specific fine-tuning. As shown in Figure 5a, classification layer fine-tuning (CFT) for `GuardFormer` outperforms full fine-tuning (FFT), whereas `PolicyGuard` requires FFT for best results. This trend is consistent across all tasks in `CustomGuardBenchmark`, with `PolicyGuard` heavily relying on FFT for generalization, particularly for "Avoid financial advice" and "Avoid Unsafe Discussion" policies. In contrast, `GuardFormer` generally achieves higher F1 scores with CFT compared to FFT, underscoring the crucial role of GIP in generalizing to novel, unseen policies.

**GuardFormer requires fewer training steps to reach optimal performance** Furthermore, `GuardFormer` demonstrates impressive data efficiency. Figure 5b illustrates that `GuardFormer` not only improves with less training data compared to `PolicyGuard` but also converges more rapidly. On average, `GuardFormer` requires just 1 epoch per task, while `PolicyGuard` needs 8. Remarkably, `GuardFormer`'s performance is nearly on par even without additional task-specific fine-tuning, indicating significantly enhanced zero-shot performance and generalization to new, unseen guardrailing policies/tasks. This zero-shot capability surpasses the baseline LLMs, highlighting the effectiveness of our GIP step on synthetic guardrail data.

## 6 Conclusion

In this work, we proposed a process for achieving highly performance discriminative classifiers that generalize well the custom policies that define the scope of a guardrail. We find that with models that are less than 512MB in storage can outperform models of magnitudes of order larger such as gpt-4 and significantly outperform well-established and publicly available guardrails such as those from the LlamaGuard suite. We view this as a breakthrough for faster and low cost guardrailing and can be used tangentially with general purpose large language models and on-device given the reduced memory and storage footprint.

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
