# OpenReview forum: "GuardFormer: Guardrail Instruction Pretraining for Efficient SafeGuarding"
_NeurIPS.cc/2024/Workshop/SafeGenAi — SafeGenAi Oral_

### Official Review · Reviewer_D7Ha · 2024-10-08
**Review for  GuardFormer: Guardrail Instruction Pretraining for Efficient SafeGuarding**

**Rating:** 8
**Confidence:** 5

**Review:**

This paper proposes Guardformer: a guardrail model that outperforms current state of the art while being much smaller than other models. The motivation and structure of paper is good, some comments below:
1. Will Guardformer face inference costs when it scales to other detectors apart from the ones tackled in paper?
2. Reliance on synthetic data could be a weakness as it does not capture real-world toxicity, safety, finance tasks. Could this limit the model's generalization ability?
3. While F1 score is a good metric to evaluate on, can the authors provide other metrics details too like precision, recall, robustness etc

---

### Official Review · Reviewer_22dC · 2024-10-09
**GuardFormer: Guardrail Instruction Pretraining for Efficient Safeguarding**

**Rating:** 8
**Confidence:** 3

**Review:**

## Summary
The paper proposes a novel method called GuardFormer for improving guardrail instruction pre-training using synthetic data generation. This method enhances large language model (LLM) safety by efficiently guarding against malicious prompts while significantly reducing inference costs and memory consumption. The authors introduce a synthetic data generation pipeline to create compliant and non-compliant prompt examples for fine-tuning models. The method is evaluated across several public and private benchmarks, showing a marked improvement in both speed and performance over existing models like GPT-4 and other publicly available solutions.

## Strengths
* Novel Approach: The proposal of using guardrail-specific instruction pretraining through synthetic data generation is innovative. The synthetic data outperforms real data in model training, and the GuardFormer models are shown to be highly efficient and scalable compared to large LLMs like GPT-4.

* Performance: GuardFormer achieves significant improvements in classification accuracy (F1 score), outperforming state-of-the-art LLMs on both public and private benchmarks. The efficiency of the model, in terms of both speed and memory consumption, makes it a promising solution for real-world applications where resource constraints are critical.

* Thorough Evaluation: The paper provides comprehensive experiments, including comparisons with various baselines such as GPT-4, Aegis-LlamaGuard, and PromptGuard. The use of multiple benchmarks, including private datasets like CustomGuardBenchmark, strengthens the reliability of the results.

* Practical Implications: GuardFormer’s ability to maintain high performance while being lightweight makes it ideal for deployment in resource-constrained environments (e.g., on-device applications). Its potential for low-cost and scalable implementation is well-argued.

## Areas for Improvement
* Clarity of Synthetic Data Generation Process: The description of the synthetic data generation process could benefit from further elaboration. While the method seems effective, the mechanics behind how synthetic data is generated, and its scalability for various domains, should be clearer, especially for readers who are unfamiliar with synthetic data generation techniques.

* Generalization Across Domains: While the paper demonstrates strong performance on specific safety tasks (e.g., financial advice, toxic behavior), it is unclear how well GuardFormer generalizes across a wider range of domains. Additional evaluation on more diverse tasks could strengthen the claim that the model performs well universally.

* Comparison with Fine-Tuned LLMs: The paper compares GuardFormer with large models like GPT-4, but it would be interesting to see how the model compares to fine-tuned, smaller models (e.g., fine-tuned versions of Llama or other instruction-following models). Including such comparisons would provide a fuller understanding of GuardFormer's efficiency.

* Ablation Study: An ablation study to analyze the impact of different components of the model (e.g., synthetic data generation vs. guardrail instruction pretraining) would help isolate which factors contribute most to the observed performance gains.

## Conclusion
The paper presents an innovative and impactful approach to guardrailing for LLMs, offering significant improvements in efficiency and performance over existing methods. Despite a few areas where more clarity or additional experimentation could be useful, the core contributions of the work are substantial and offer a promising solution for the future of LLM safety.